# Evaluation of Implant Stability and Trephination Depth for Implant Removal—An In Vitro Study

**DOI:** 10.3390/ma15124200

**Published:** 2022-06-13

**Authors:** Haya Meir, Alon Sebaoun, Perry Raz, Shifra Levartovsky, Adi Arieli, Raphel Pilo, Zafar Dor, Ilan Beitlitum

**Affiliations:** 1Department of Periodontology and Dental Implants, The Maurice and Gabriela Goldschleger School of Dental Medicine, The Sackler Faculty of Medicine, Tel Aviv University, Tel Aviv 6997801, Israel; hayameir@012.net.il (H.M.); alon.sebaoun@gmail.com (A.S.); drperioraz@gmail.com (P.R.); 2Department of Oral Rehabilitation, The Maurice and Gabriela Goldschleger School of Dental Medicine, The Sackler Faculty of Medicine, Tel Aviv University, Tel Aviv 6997801, Israel; shifralevartov@gmail.com (S.L.); dr.arieli@gmail.com (A.A.); 3Department of Oral Biology, The Maurice and Gabriela Goldschleger School of Dental Medicine, The Sackler Faculty of Medicine, Tel Aviv University, Tel Aviv 6997801, Israel; rafipilo@gmail.com; 4The Maurice and Gabriela Goldschleger School of Dental Medicine, The Sackler Faculty of Medicine, Tel Aviv University, Tel Aviv 6997801, Israel; zafardor@gmail.com

**Keywords:** trephination, implant removal, ISQ

## Abstract

Malpositioned and broken implants are usually fully osseointegrated; hence, their removal, especially from the lower arch, can be very challenging. Implant removal techniques include reverse torque and trephination. Trephination is an invasive technique that can jeopardize vital structures, cause mandibular fatigue fractures, or lead to osteomyelitis. In this study, we aimed to assess the relationship between trephination depth and implant stability by recording implant stability quotient (ISQ) readings at varying trephination depths in vitro. Materials and methods: Forty-eight implants were inserted into dense synthetic polyurethane foam blocks as artificial bone. Primary implant stability was measured with a Penguin resonance frequency analysis (RFA) device. Implants of two designs with a diameter of 3.75 mm and a length of 13 or 8 mm were inserted. Twenty-four internal hexagon (IH) (Seven^®^) and twenty-four conical connection (CC) implants (C1^®^; MIS^®^ Implants, Ltd., Misgav, Israel) were used. The primary implant stability was measured with the RFA device. Trephination was performed, and implant stability was recorded at depths of 0, 3, and 6 mm for the 8 mm implants and 0, 3, 6, 8, 10, and 11.5 mm for the 13 mm implants. Results: Linear regression revealed a significant relation between the trephination depth and the ISQ (F (1, 213) = 1113.192, *p* < 0.001, adjusted r2 = 0.839). The trephination depth significantly predicted the ISQ (β = −5.337, *p* < 0.001), and the ISQ decreased by −5.33 as the trephination depth increased by 1 mm. Conclusion: Implant stability reduction as measured using an RFA device during trephination may be a valuable guide to achieving safe reverse torque for implant removal. Further studies are needed to evaluate these data in clinical settings.

## 1. Introduction

The main reasons for implant removal are bone loss caused by peri-implantitis or mechanical complications, such as implant platform or screw fractures [1,2,3]. In some cases, malpositioned implants may also need to be removed. Fully osseointegrated malpositioned or broken implants are difficult to remove, especially in the lower arch [4,5]. Moreover, neighboring teeth and anatomical structures may be unintentionally harmed in the process [6].

The most common methods used for implant removal are reverse torque and trephination. Reverse torque should be the first choice despite its inferior success rate [5] since it is more conservative, less invasive and less traumatic to the surrounding bone and the patient than trephination. Reverse torque can be effective yet challenging when the implant is surrounded by dense bone. In cases in which only softer bone surrounds the implant, as is commonly seen in the anterior region of the mouth, removal via the reverse torque technique may leave little or no facial bone for future implant placement. Reverse torque as a standalone technique in the mandible, especially for the removal of fully integrated implants, is unpredictable: failure can lead to either a broken reverse torque retrieval tool inside the implant or a broken implant head [4,5]. Trephination is a more controlled technique for implant removal but requires flap elevation. Moreover, to visualize the position and angulation of the implant to properly position the trephine drill, one may need to remove the cervical peri-implant bone [4,5]. Trephination is an invasive technique that can cause mandibular fatigue fractures and osteomyelitis [3,7].

The implant surface type may influence the success rate of the reverse torque technique for the removal of screw-type implants [6]; however, the quality of the bone and, therefore, the implant stability as measured by a resonance frequency analysis (RFA) device are more crucial parameters.

Four millimeters of remaining osseointegration was described as a critical threshold for successfully removing an implant using the reverse torque method alone [8].

A combined approach is especially important in cases where the implant is in close proximity to vital anatomical structures, such as the mental nerve or mandibular canal [2,9,10]. However, the effective drilling depth of trephination to allow for the safe application of the countertorque ratchet remains unclear.

In this study, we aimed to assess the relationship between the trephination depth and implant stability by recording implant stability quotient (ISQ) readings at varying trephination depths in vitro. An ISQ measurement during trephination can guide the clinician to a point where the reverse torque technique can be safely and predictably applied to remove an implant.

## 2. Materials and Methods

The sample size was calculated using G*power Software (version. 3.1.9.7, University of Heinrich-Heine, Düsseldorf, Germany), with α = 5% and a power (1-β) of 80%. Based on the values obtained from a previous study [11] the standard deviation of the ISQ values, applying the same measuring device, were 6.62.

Our assumption was that differences of one standard deviation in ISQ between two groups are clinically relevant, thus the sample size was estimated to be 34 (17 in each group). Finally, our sample consisted of 48 implants (24 in each group).

Since no previous data was available concerning the correlation calculation, in post hoc analysis using α = 0.05, adjusted r2 = 0.839, for a sample size of 48, the power (1-β) was larger than 80%.

Forty-eight implants were inserted into an artificial bone material made of dense synthetic polyurethane foam blocks 120 × 170 × 42 mm in size (Sawbones, Malmö, Sweden). These dense bone blocks (#40) were characterized by a density of 0.64 gr/cc and laminated on one side by 2 mm of cortical bone (#50) with different cortical thicknesses and trabecular densities. The cortical thickness of the mandible was 2.22 mm. Our block density was increased to examine extreme conditions. Implants of two designs with a diameter of 3.75 mm were inserted. The first was a tapered internal hexagon (IH) implant (Seven^®^, MIS^®^ Implants, Ltd., Misgav, Israel) and the second was less tapered with a conical connection (CC; C1^®^, MIS^®^ Implants, Ltd., Misgav, Israel). The characteristics of the two implant designs were as follows: The IH implant was characterized by a tapered design, with an interthread distance of 2 mm, while the CC implant was less tapered, with an interthread distance of 1.5 mm; thus, there are more threads in the latter implant. The CC implant had two spiral channels in its apex, while the IH implant had three channels. The threads in the IH implant are deeper than those in the CC implant. In both implants, the different thread designs condense at the neck and cut at the apex, which is domed; both implants also had a platform-switching microgap. There were four experimental groups, with 12 implants in each group and two different lengths.

Group 1: (CC) 12 implants 13 mm in length.

Group 2: (IH) 12 implants 13 mm in length.

Group 3: (CC) 12 implants 8 mm in length.

Group 4: (IH) 12 implants 8 mm in length.

The implants were inserted at a constant distance of 30 mm from each other over the block.

Primary implant stability was measured with a Penguin RFA device (Penguin Integration Diagnostics, Göteborg, Sweden) after the transducer (Smartpeg) was screwed to the implant. Three repeated measurements from different directions were obtained for each implant, and the average number was recorded.

Trephination was performed using a trephine with an internal diameter of 5 mm and an external diameter of 6 mm (MIS^®^ Implants, Ltd., Israel). The ISQ was recorded during trephination at depths of 0, 3, and 6 mm for the 8 mm implants and 0, 3, 6, 8, 10, and 11.5 mm for the 13 mm implants (Figure 1).

### Statistical Analysis

To evaluate the correlation between the ISQ and trephination depth and compare them by implant type (CC vs. IH) or implant length (13 mm vs. 8 mm), t-tests for independent samples were used. Pearson’s correlation test was used to evaluate these correlations. The ISQ values are expressed as the mean and standard deviation (SD). *p* < 0.05 was considered significant. Statistical analysis was performed using SPSS Software (version 25, IBM, New York, NY, USA).

## 3. Results

The mean ISQ values are shown in Table 1. The implant type, mean ISQ, and trephination depth are presented in Figure 2.

There was a significant difference in the ISQs between the 8 mm and 13 mm implants (*t* = 2.921, *p* = 0.004), with a mean difference of 9.45. However, the t-test showed no significant difference in the ISQs between the implant designs (*t* = 0.531, *p* = 0.596).

Therefore the two implant designs were united. Figure 3 showed the ISQ at each trephination depth in the unified group; in general, there was a negative linear correlation between the trephination depth and the ISQ (r = −0.916, *p* < 0.001).

When considering the ISQ as a function of trephination depth (Figure 3), linear regression revealed a significant relationship between the trephination depth and the ISQ (F (1, 213) = 1113.192, *p* < 0.001, adjusted r2 = 0.839). The trephination depth significantly predicted the ISQ (β = −5.337, *p* < 0.001), and the ISQ decreased by −5.33 as the trephination depth increased by 1 mm. The predictive model was as follows:
Implant stability = (initial ISQ) − 5.337 * (trephination depth).

After controlling for the implant length, this correlation was still significant (r = −0.936, *p* < 0.001).

## 4. Discussion

The implant design did not influence the ISQ during trephination, and the t-tests showed no significant differences in implant stability between the two implant designs (*t* = 0.531, *p* = 0.596); therefore, we united the two designs (CC and IH) into one group and analyzed the results accordingly.

In a previous article, we demonstrated the reliability and ease of use of the Penguin RFA device for ISQ measurements [11].

Only dense foam blocks were chosen in this experiment since they better simulate the clinical setting of dense bone, particularly in the lower arch, where real difficulties concerning osseointegrated implant removal are faced.

The mean ISQ was 76.29 (SD 8.14) for the 13 mm implants. These results are in agreement with those of a previous study on RFA and bone loss during the early healing period in humans [12]. The high ISQ values were related to the bone density, which resulted in high primary stability.

The ISQ decreased by −5.33 as the trephination depth increased by 1 mm, with no effect from implant length or design.

Increasing the safety and reducing the failure rate of the reverse torque technique for implant removal is very important; hence, clinicians need a reliable quantitative measurement during this stressful procedure. During trephination, there is a decrease in implant stability, as indicated by the ISQ measured using RFA. For the 13 mm implants, we noticed that 6–8 mm trephination was a critical range in which the ISQ (54–41) decreased dramatically. We assume that this was the cut-off value of implant stability to be applied in order to predictably switch from trephination to reverse torque in dense bone.

To the best of our knowledge, this is the first report of this linear correlation between trephination depth and implant stability as evaluated using RFA; hence, this model represents the clinical potential of the continual evaluation of implant stability during trephination in dense bone with an RFA device. However, this concept needs further support from clinical and animal studies. Implant removal techniques described in the literature include the use of a countertorque ratchet, piezosurgery, high-speed burs, elevators, forceps, trephine burs, and laser surgery [3]. One can facilitate reverse torque implant removal by heating the implant. Thermal necrosis-aided implant removal has also been proposed as a minimally invasive method, and this technique has been evaluated using 3D finite element analysis [4] and in porcine cadaver and animal studies [5,7]. The application of this promising technique in humans has yet to be validated, and the appropriate clinical settings for this procedure should be defined [6,7]. Limited data concerning implant thermoremoval in humans are currently available [13,14,15]. To the best of our knowledge, only one case report describing the usage of electrosurgery for the simple removal of poorly positioned fixtures in the premaxilla is available [16]. The application of a temperature over 47 °C for 1 min was shown to induce “heat necrosis” [13]. The risk and extent of the damage to the surrounding bone are unpredictable, and data addressing patient discomfort and related pain during this stage are lacking.

The need for combinations of such techniques usually arises in dense bone in the mandible.

The implant surface type may influence the success rate of the reverse torque technique for the removal of screw-type implants [6]; however, the quality of the bone and, therefore, the implant stability, as indicated by the ISQ measured using RFA, are more crucial parameters.

The maintenance and preservation of hard tissue and subsequent site development should always be considered in the implant removal phase since reimplantation at the site is usually needed. Imprecise, traumatic removal of implants may leave large bony defects that impair reimplantation possibilities.

Clinicians may use trephination-based, guided implant surgery for implant placement [17] and removal [18]. A few studies reported the use of trephine drills for implant site preparation [19]. This technique may reduce thermal bone damage and may simplify the drilling protocol [12,17]. However, using solely trephine drills for implant removal, especially in a malpositioned implant, can jeopardize new implant placement since the trephine diameter used is wider than the original implant diameter.

The use of an RFA device during trephination may serve as a valuable guide in combined implant removal approaches. An ISQ reading that designates sufficient implant stability reduction via trephination to allow for safe reverse torque application is crucial to minimize bone damage via trephination and maximize the efficacy and safety of implant removal using reverse torque. This in vitro study had its limitations. In a clinical setting, we would have chosen a trephine bur with an internal diameter of 4 mm for the removal of 3.75 mm diameter implants. In our model, we were obliged to use a wider bur (5 mm) since the bur was in close contact with the implant and removed it at all depths of trephination.

This model represents mechanical implant stability but does not simulate implant osseointegration. We were able to isolate the effect of trephination depth on implant stability; however, we could not apply reverse torque implant removal, which is the second part of the procedure.

The purpose of this study was to validate the principle of coupling trephination and ISQ, which is an objective parameter for assessing implant stability regardless of osseointegration. As of today, there are no objective clinical guidelines for implant removal; therefore, based on our study, we suggest using the ISQ parameter as a useful indicator for applying a minimal yet effective force for safe implant removal. Further studies are needed to evaluate these data in clinical settings.

## Figures and Tables

**Figure 1 materials-15-04200-f001:**
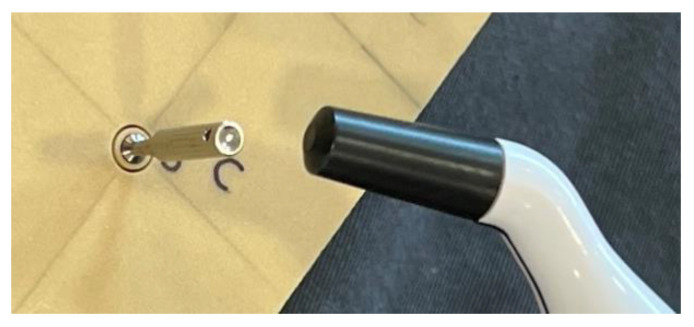
Representative photo of how the measurement were performed.

**Figure 2 materials-15-04200-f002:**
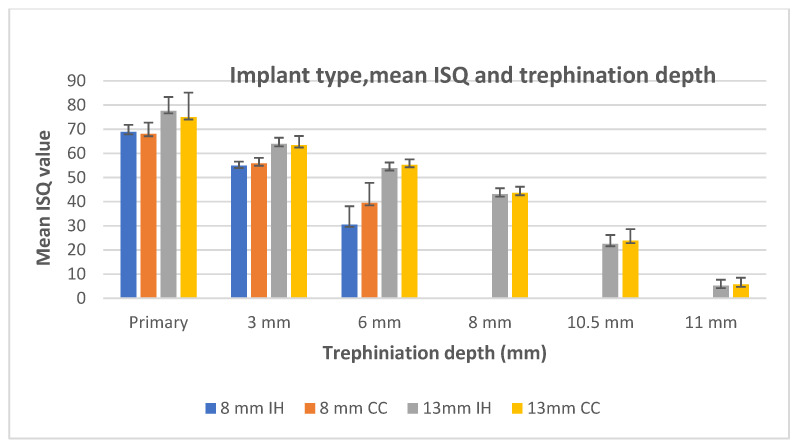
Implant type, mean ISQ, and trephination depth.

**Figure 3 materials-15-04200-f003:**
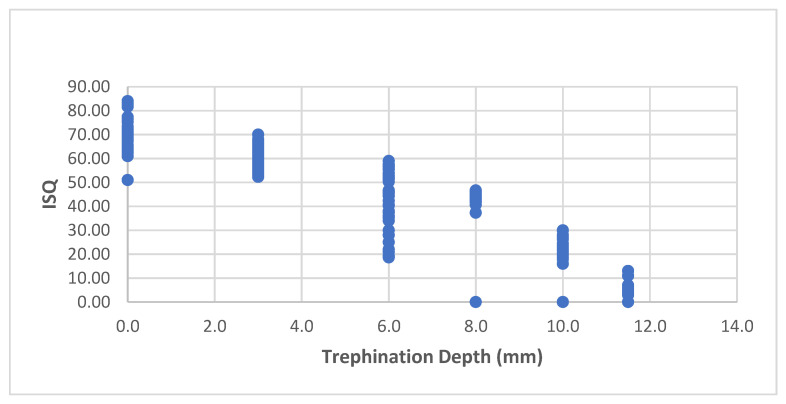
ISQ at each trephination depth in the united group.

**Table 1 materials-15-04200-t001:** The mean ISQ values and trephination depths.

Implant Length	Trephination Depth (mm)	N	Mean	SD
8 mm	0	24	68.48	3.83
	3	24	55.42	1.98
	6	23	35.20	9.02
13 mm	0	24	76.29	8.14
	3	24	63.68	3.14
	6	24	54.59	2.32
	8	24	41.58	9.18
	10	24	22.26	6.30
	11.5	24	4.58	3.15

## Data Availability

Data presented in this in this study are available on request from the corresponding author.

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
