# Peer review of "Evaluation of Implant Stability and Trephination Depth for Implant Removal—An In Vitro Study"

_materials, 2022, doi:10.3390/ma15124200_

Round 1

Reviewer 1 Report

In this study, the researchers studied the relationship between trephination depth and implant stability by recording implant stability quotient (ISQ) readings at varying trephination depths in vitro. This is an interesting study to access implant stability. Some revisions are requested.

Abstract

Line 15. Remove the word “Introduction”.

Please follow the Journal guidelines for the referencing.

Line 39-41. Please modify the sentence. The malposition implants are not always osseointegrated. It is better to not put this meaning. It can be stated as… Fully osseointegrated malposition implants are difficult to remove.

Line 57. Remove the word “very”.

Discussion.

Can the trephination be affected by bone angulation? And bone angulation can affect the implant position. Please discuss the bone angulation with implant placement.  https://www.mdpi.com/1996-1944/15/9/3004

Please discuss using the following related articles

https://pubmed.ncbi.nlm.nih.gov/30503150/

https://meridian.allenpress.com/joi/article-abstract/47/3/199/442396/In-Vitro-Comparison-of-Time-and-Accuracy-of?redirectedFrom=fulltext

Author Response

The manuscript entitled “Evaluation of implant stability and trephination depth for Im-plant removal - an in vitro study” submitted to Materials

We take the opportunity to thank the reviewer for this in lightning remarks that helped us ameliorate our work.

Abstract

Line 15. Remove the word “Introduction”.

Was removed 

Please follow the Journal guidelines for the referencing.

The manuscript was revised according to the journal guidelines using the specific journal template. Line 39-41. Please modify the sentence. The malposition implants are not always osseointegrated. It is better to not put this meaning. It can be stated as… Fully osseointegrated malposition implants are difficult to remove.

Was modified

Line 57. Remove the word “very”.

The word was removed

Discussion.

Can the trephination be affected by bone angulation? And bone angulation can affect the implant position. Please discuss the bone angulation with implant placement.  https://www.mdpi.com/1996-1944/15/9/3004

Please discuss using the following related articles

https://pubmed.ncbi.nlm.nih.gov/30503150/

https://meridian.allenpress.com/joi/article-abstract/47/3/199/442396/In-Vitro-Comparison-of-

Time-and-Accuracy-of?redirectedFrom=fulltext

Thank you for raising this important issue, a new paragraph was added to the discussion

Clinicians may use trephination-based, guided implant surgery for implant placement(Suriyan et al. 2019) and removal(Deeb et al. 2018). A few studies reported the use of trephine drills for implant-site preparation(Deeb et al. 2021). This technique may reduce thermal bone damage and may simplifies the drilling protocol(Stelzle et al. 2014)(Suriyan et al. 2019). However, using solely trephine drills for implant re-moval especially in malpositioned implant can jeopardize new implant placement since the trephine diameter used is wider than the original implant diameter.

Reviewer 2 Report

The authors speak of four experimental groups of 12 implants each. They should define the four groups because it is confusing and confusing. In addition, in Table 1 in 8 mm implants and 6 mm trepanation, the N is 23, because it is not 24.

I advise adding a clinical figure in which images with different trepanations are seen.

Author Response

The manuscript entitled “Evaluation of implant stability and trephination depth for Implant removal - an in vitro study” submitted to Materials

We take the opportunity to thank the reviewer for this important  remarks that helped us ameliorate our work .

The authors speak of four experimental groups of 12 implants each. They should define the four groups because it is confusing and confusing. In addition, in Table 1 in 8 mm implants and 6 mm trepanation, the N is 23, because it is not 24

We add a new paragraph.

There were 4 experimental groups, with 12 implants in each group and two different lengths.

Group 2, (IH) 12 implants 13 mm length.

Group 3, (CC) 12 implants 8 mm length.

Group 4, (IH) 12 implants 8 mm length.

One implant was lost .

I advise adding a clinical figure in which images with different trepanations are seen.

An illustrative fig. 3 was added   

Reviewer 3 Report

The aim of the present study was to assess the relationship between trephination depth and implant stability by recording implant stability quotient (ISQ) readings at varying trephination depths in vitro. It is not clear the reason why the authors want to conduct those experiments; is it to validate a protocol for the combined use of reverse torque and trephination in order to find a balance betwen the advantages and disadvantages of the two techniques? 

Abstract: Please, spell out at first use "MIS implants". Please, specify which are the two implants' designs. 

Introduction: Maybe the authors mean "dense" bone instead of "thick" bone (lines 47-48). Why the issue of bone thickness is mentioned only regarding reverse torque and not trephination?

Methods: Sample size calculation is missing and it crucial for the validity of the conducted analyses. Also, it is not specified how many CC and IH samples have been used in each experimental group? Why have implants with a different design been chosen? 

Discussion: the first sentence can be deleted as it repeats the aim and the results. It is not clear why in the discussion the authors introduced the topic of implant thermoremoval instead of commenting on their results. From a clinical point of view and to enhance the traslational potential of this work it would have been useful to find a cut off value of implant stability to be used to predicatbly switch from trephination to reverse torque. 

To my opinion, the results of present are more feasible for a implantology journal. 

Author Response

The manuscript entitled “Evaluation of implant stability and trephination depth for Im-plant removal - an in vitro study” submitted to Materials

We take the opportunity to thank the reviewer for this most constructing remarks that helped us ameliorate our work .

Abstract: Please, spell out at first use "MIS implants". Please, specify which are the two implants' designs.

A modified paragraph was added to the abstract .

Twenty-four internal hexagon (IH) (Seven®) and twenty-four conical connection implants (CC; C1®) ;MIS® Implants, Ltd., Israel) 

Introduction: Maybe the authors mean "dense" bone instead of "thick" bone (lines 47-48). Why the issue of bone thickness is mentioned only regarding reverse torque and not trephination?

Correct, this was modified

Methods: Sample size calculation is missing and it crucial for the validity of the conducted analyses. Also, it is not specified how many CC and IH samples have been used in each experimental group? Why have implants with a different design been chosen?

Sample size was determent by statistical analysis (G*power3.1.9.4)  

We assume that implant design have different primary implant stability, however in dense bone, Implant design did not influence the ISQ during trephination

Discussion: the first sentence can be deleted as it repeats the aim and the results. It is not clear why in the discussion the authors introduced the topic of implant thermoremoval instead of commenting on their results. From a clinical point of view and to enhance the traslational potential of this work it would have been useful to find a cut off value of implant stability to be used to predicatbly switch from trephination to reverse torque. 

This sentence was deleted and the relevant paragraph was modified.

We noticed that 6-8 mm trephination was a critical range in which the ISQ (54-41) de-creased dramatically. We assume that this is the cut off value of implant stability to be applied in order to predictably switch from trephination to reverse torque in dense bone.

This part in the discussion dealt with different conservative technique for implant removal.  We think it was important to mention this innovative technique.  

To my opinion, the results of present are more feasible for a implantology journal. 

Although we agree with this comment, we think that this type of manuscript is relevant to this materials' special issue.

Round 2

Reviewer 1 Report

The authors have addressed the comments and the manuscript is improved. It can be accepted but minor formatting is needed.

Author Response

The authors have addressed the comments and the manuscript is improved. It can be accepted but minor formatting is needed.

Minor formatting was made. and final graphical editorial changes are expected.    

Reviewer 3 Report

The article has slightly improved, although my ratings are not changed. It is unclear how the sample size was calculated and it should be detailed in the materials and methods section of the article.  Figure 3 is quoted before Figs. 1 and 2. The figures should be quoted in the order of appearance in the text. 

Author Response

The article has slightly improved, although my ratings are not changed. It is unclear how the sample size was calculated and it should be detailed in the materials and methods section of the article.  Figure 3 is quoted before Figs. 1 and 2. The figures should be quoted in the order of appearance in the text. 

A new paragraph was added 

The sample size was calculated using G*power 3.1.9.4 software, with an α=5%, and the power (1-β) of 80%. Based on the values obtained from a  previous study (Raz et al. 2021), the standard deviation of the ISQ values, applying the same measuring device were 6.62.

Our assumption was that differences of one standard deviation in ISQ between two groups are clinically relevant, thus the sample size was estimated to be 34. Finally, our sample consisted of 48 implants (24 in each group).

Since no previous data was available concerning the correlation calculation, in post hoc analysis using α=0.05, adjusted r2=0.839, for a sample size of 48, the power (1-β) was larger than 80%.

We changed the Figure numbers and they are quoted in the order of appearance in the text.